# Appropriate Nitrogen form Ratio and UV-A Supplementation Increased Quality and Production in Purple Lettuce (*Lactuca sativa* L.)

**DOI:** 10.3390/ijms242316791

**Published:** 2023-11-27

**Authors:** Binbin Liu, Pengpeng Mao, Qi Yang, Hengshan Qin, Yaliang Xu, Yinjian Zheng, Qingming Li

**Affiliations:** 1College of Food and Biological Engineering, Chengdu University, Chengdu 610106, China; lbroom@163.com; 2Institute of Urban Agriculture, Chinese Academy of Agricultural Sciences, Chengdu 610218, China; mao2peng@126.com (P.M.); xuyaliang@caas.cn (Y.X.); 3College of Agriculture, Nanjing Agricultural University, Nanjing 210095, China; 4College of Agronomy and Biotechnology, Southwest University, Chongqing 400716, China; canshuizhi3@163.com; 5College of Horticulture Science and Engineering, Shandong Agricultural University, Tai’an 271018, China; qhs1021217321@163.com

**Keywords:** anthocyanin, nitrogen form, nitrate, purple lettuce, ultraviolet-A

## Abstract

Purple lettuce (*Lactuca sativa* L. cv. Zhongshu Purple Lettuce) was chosen as the trial material, and LED intelligent light control consoles were used as the light sources. The purpose was to increase the yield and quality of purple lettuce while lowering its nitrate level. By adding various ratios of NO_3_^−^-N and NH_4_^+^-N to the nutrient solution and 20 µmol m^−2^ s^−1^ UV-A based on white, red, and blue light (130, 120, 30 µmol m^−2^ s^−1^), the effects of different NO_3_^−^-N/NH_4_^+^-N ratios (NO_3_^−^-N, NO_3_^−^-N/NH_4_^+^-N = 3/1, NH_4_^+^-N) and UV-A interaction on yield, quality, photosynthetic characteristics, anthocyanins, and nitrogen assimilation of purple lettuce were studied. In order to produce purple lettuce hydroponically under controlled environmental conditions, a theoretical foundation and technological specifications were developed, taking into account an appropriate UV-A dose and NO_3_^−^-N/NH_4_^+^-N ratio. Results demonstrate that adding a 20 µmol m^−2^ s^−1^ UV-A, and a NO_3_^−^-N/NH_4_^+^-N treatment of 3/1, significantly reduced the nitrate level while increasing the growth, photosynthetic rate, chlorophyll, carotenoid, and anthocyanin content of purple lettuce. The purple leaf lettuce leaves have an enhanced capacity to absorb nitrogen. Furthermore, plants have an acceleration of nitrogen metabolism, which raises the concentration of free amino acids and soluble proteins and promotes biomass synthesis. Thus, based on the NO_3_^−^-N/NH_4_^+^-N (3/1) treatment, adding 20 µmol m^−2^ s^−1^ UV-A will be helpful in boosting purple lettuce production and decreasing its nitrate content.

## 1. Introduction

Because of controlled-environment cultivation and a large range of generated varieties, lettuce is one of the most popular leafy vegetables consumed globally and is available all year [1]. Furthermore, it is a rich source of natural phytochemicals and nutritional bioactive substances such as sesquiterpene lactones, glycosylated flavonoids, phenolic acids, carotenoids, vitamin B groups, ascorbic acid, and tocopherols, which may have anti-free radical, anti-inflammatory, anti-cancer, and anti-diabetes effects [1,2]. Nitrogen and light supplementation are critical in modern crop production for increasing yield and quality. Numerous prior studies have shown that nitrogen and light quality have varied impacts on lettuce growth and development, nitrogen uptake and utilization, phytochemical content and accumulation, and photosynthetic product allocation and utilization [3,4,5,6,7,8]. However, the interaction effects of nitrogen supply and light quality on plant growth and development, photosynthetic rate, nitrate concentration, and enzyme activity have received less attention.

Nitrogen, a vital macronutrient for plants, is also an important component of phospholipids, chloroplasts, nucleic acids, and proteins [9,10]. Nitrogen metabolism-related enzymes such as nitrate reductase, nitrite reductase, glutamine synthetase, glutamate synthase, and glutamate dehydrogenase all have an immediate impact on amino acid synthesis and transformation. Plant nitrogen absorption, as well as utilization, are influenced not only by their own physiological properties but also by the nitrogen source available in their growth environment [11]. Nitrate nitrogen (NO_3_^−^-N) and ammonium nitrogen (NH_4_^+^-N) are the two primary types of inorganic nitrogen taken and utilized by plants [8]. Different nitrogen types can influence a plant’s physiological processes by changing the ion balance in vivo, carbohydrate, and other mechanisms [12]. According to previous research findings, vegetable soybeans grow and develop better when the suitable ammonium nitrate nitrogen to nutrient solution ratio is 75:25; however, it is unclear in lettuce. Furthermore, owing to increased nitrogen fertilizer use, leafy plants like lettuce have a high nitrate content, which can lead to harmful nitrosamine conversion and food safety issues [8,10,13]. As a result, it is critical for lettuce production to keep nitrate concentrations within an acceptable range [10].

Blue and red light are mostly applied in protected horticulture, whereas other light qualities are typically lacking, including ultraviolet-A (UV-A), which is one of the environmental elements that can penetrate the ozone layer and clouds to reach the earth’s surface under natural conditions [3,14,15,16,17]. Results from earlier studies demonstrate how UV-A affects the accumulation of biomass, the distribution of resources, the shape of leaves, photosynthesis, and the biochemical production and content of lettuce and other vegetables [17,18,19]. A large decrease in the content of nitrate in lettuce may be caused by using the broad spectral energy of red and blue light in mixed red, blue, and white light [3]. Furthermore, when applied in increasing amounts, pre-harvest nitrogen restriction and continuous lighting improve quality and flavor, lower nitrate levels, and impact lettuce growth and ASA metabolism [9,14,20,21]. However, the interaction of UV-A supplementation and nitrogen supply on lettuce is unknown. As a result, this experiment explored the relationship between UV-A and nitrogen forms in hydroponic purple lettuce by evaluating the growth and quality of hydroponic purple lettuce under different nitrogen forms and UV-A. A theoretical foundation and technological criteria are provided by hydroponic purple lettuce.

## 2. Results

### 2.1. Effects of Different Nitrogen forms and UV-A Interaction on the Growth of Purple Lettuce

Table 1 shows that, in the absence of UV-A, the NO_3_^−^-N/NH_4_^+^-N (3/1) treatment grew significantly more than the NO_3_^−^-N and NH_4_^+^-N treatments; the NH_4_^+^-N treatment had the lowest leaf area and dry and fresh weight, coming in at 78.27%, 81.72%, and 67.15% less than the NO_3_^−^-N/NH_4_^+^-N (3/1) treatment. In the presence of UV-A, the growth of the NO_3_^−^-N/NH_4_^+^-N (3/1) treatment was significantly higher than that of single nitrate and ammonium nitrogen, the lowest NH_4_^+^-N treatment. In comparison to the NO_3_^−^-N/NH_4_^+^-N (3/1), the leaf area, fresh weight, and dry weight decreased by 70.16%, 79.32%, and 63.64%, respectively; the length of stems, stem thickness, leaf area, and dry and fresh weight of different nitrogen forms after adding UV-A were higher than those of different nitrogen forms without UV-A. From the F-values analysis, we found that the interaction effect of nitrogen forms and UV-A supplementation on purple lettuce morphology is mainly reflected in stem length, and nitrogen forms exhibited significant effects on plant growth. This demonstrates that the addition of UV-A makes appropriate ammonium nitrate ratio treatment more conducive to increasing hydroponic purple lettuce yields. Nevertheless, hydroponically grown purple lettuce did not grow well when treated with NH_4_^+^-N.

### 2.2. Effects of Different Nitrogen Forms and UV-A Interactions on Photosynthetic Pigment Content of Purple Lettuce

In the different nitrogen forms after adding UV-A, the chlorophyll a, chlorophyll b, chlorophyll a + b, and carotenoids were significantly higher than the different nitrogen forms without UV-A. As shown in Table 2, the treatment with NO_3_^−^-N/NH_4_^+^-N (3/1) was significantly higher than that of the total nitrate and total ammonium treatments. The NO_3_^−^-N treatment was significantly higher than other treatments, with the exception of chlorophyll a/b, and differences between other treatments were not statistically significant. Intriguingly, for the photosynthetic pigment content of purple lettuce, neither the nitrogen forms nor the UV-A supplementation exhibited an interaction effect. This demonstrated that the addition of 20 µmol m^−2^ s^−1^ UV-A and the appropriate NO_3_^−^-N/NH_4_^+^-N (3/1) led to an improvement in the chlorophylls a, b, a + b, as well as in the carotenoids in hydroponically grown purple lettuce.

### 2.3. Effects of Different Nitrogen Forms and UV-A Interactions on Gas Exchange Parameters of Purple Lettuce

Under the NO_3_^−^-N/NH_4_^+^-N (3/1) treatment, the net photosynthetic rate of hydroponically grown purple lettuce was higher than that of other treatments. The net photosynthetic rate under the NO_3_^−^-N/NH_4_^+^-N (3/1) treatment was not statistically significance whether or not UV-A was applied, and among them, the NH_4_^+^-N treatment with 20 µmol·m^−2^·s^−1^ UV-A was showed the lowest net photosynthetic rate (Figure 1). After adding UV-A, the NO_3_^−^-N/NH_4_^+^-N (3/1) treatment had increased stomatal conductance, transpiration rate, and intercellular CO_2_ concentration compared to the other treatments, but the NH_4_^+^-N treatment showed no appreciable difference.

### 2.4. Effects of Different Nitrogen Forms and UV-A Interactions on the Chlorophyll Fluorescence Parameters of Purple Lettuce Leaves

It can be seen from Table 3 that, except for the dark adaptation PSII maximum quantum yield (*F*_v_/*F*_m_) under the UV-A and NO_3_^−^-N/NH_4_^+^-N (3/1) treatments, the difference between other treatments was not significant; there was no difference in the maximum quantum yield (*F*_v_′/*F*_m_′) of PSII under each light adaptation. The change trend of the actual photochemical efficiency (*Φ*_PSII_) of PSII under light adaptation was consistent with the trend of electron transfer rate (ETR) and the highest value showed under NO_3_^−^-N/NH_4_^+^-N (3/1) treatment. But, on the whole, the treatment without UV-A was higher than the treatment with UV-A, and the photochemical quenching coefficient (*q*_P_) was the same as above. The interaction effect of nitrogen forms and UV-A supplementation on chlorophyll fluorescence parameters is mainly reflected in NPQ. This shows that the ratio of the NO_3_^−^-N/NH_4_^+^-N (3/1) treatment is more conducive to improving the open ratio of PSII reaction centers under light adaptation than single nitrate nitrogen or ammonium nitrogen treatment, enhances PSII center activity, and promotes electron transfer rate.

### 2.5. Effects of Different Nitrogen Forms and UV-A Interactions on Quality of Purple Lettuce

There is no difference in the soluble sugar content among the treatments except for the NO_3_^−^-N treatment without the addition of UV-A (Table 4). The soluble protein content was highest under the NH_4_^+^-N treatment; in terms of different nitrogen forms, the content of free amino acids and vitamin C was the highest under the treatment of NO_3_^−^-N/NH_4_^+^-N (3/1). After the addition of UV-A, the content of free amino acids and vitamin C under the treatment of NO_3_^−^-N/NH_4_^+^-N (3/1) were higher than that of the treatment without the addition of UV-A NO_3_^−^-N/NH_4_^+^-N (3/1), which increased by 0.43%, 4.03%; nitrate content increased with the increase in nitrate nitrogen content, but no matter whether UV-A was added or not, it was highest under the NO_3_^−^-N treatment and lowest under the NH_4_^+^-N treatment. The increase in UV-A strength reduces the content of nitrate, which indicates that the addition of UV-A can reduce the content of nitrate.

### 2.6. Effects of Different Nitrogen Forms and UV-A Interactions on Secondary Metabolites of Purple Lettuce

The contents of total phenols, flavonoids, and anthocyanins in hydroponic purple lettuce under different nitrogen forms added with UV-A were higher than those treated with different nitrogen forms without UV-A, and the total phenol and flavonoids increased with the increase in the ratio of NH_4_^+^-N. The total phenol content after the NO_3_^−^-N/NH_4_^+^-N (3/1) treatment with UV-A was significantly higher than that found after the NO_3_^−^-N/NH_4_^+^-N (3/1) treatment without UV-A, which increased by 93.41%. The anthocyanin content in the NO_3_^−^-N/NH_4_^+^-N (3/1) treatment with UV-A was significantly higher than the other treatments (Figure 2), which were, respectively, higher than the NO_3_^−^-N and NH_4_^+^-N of UV-A processing at 53.84% and 47.49% higher. This shows that the addition of 20 µmol m^−2^ s^−1^ UV-A will promote the total phenol, flavonoids, and anthocyanins in hydroponic purple lettuce with different nitrogen forms; the anthocyanin content was the best when the UV-A was added to the NO_3_^−^-N/NH_4_^+^-N (3/1) treatment.

### 2.7. Effects of Different Ratios of Ammonium Nitrate and UV-A on Nitrate Nitrogen Content and Its Reduction in Purple Lettuce

The content of nitrate nitrogen (NO_3_^−^-N), nitrate reductase (NR), and nitrite reductase (NiR) after adding UV-A were higher than the treatment of nitrogen forms without adding UV-A, and the difference was more significant (Figure 3). From nitrogen forms, the content of NO_3_^−^-N was the lowest under the NH_4_^+^-N treatment, while the NiR and NO_3_^−^-N content was the highest under the NO_3_^−^-N/NH_4_^+^-N (3/1) treatment; after the addition of UV-A, the NR content under the treatment of NO_3_^−^-N, NO_3_^−^-N/NH_4_^+^-N (3/1) and NH_4_^+^-N was higher than each nitrogen form treatment without the addition of UV-A, which increased by 51.01%, 72.67%, and 29.50%, respectively.

### 2.8. Effects of Different Ratios of Ammonium Nitrate and UV-A on Ammonium Nitrogen Content and Assimilation of Purple Lettuce

After the addition of UV-A, the activity of glutamate synthetase (GOGAT) and glutamate dehydrogenase (GDH) are higher than those without UV-A, and GDH activity is particularly significant; after the addition of 20 µmol m^−2^ s^−1^ UV-A, the treatments of NO_3_^−^-N, NO_3_^−^-N/NH_4_^+^-N (3/1) and NH_4_^+^-N were 82.74%, 260.48%, and 17.08% higher than those of the nitrogen forms without UV-A treatment, respectively. While the activity of glutamine synthetase (GS) was contrary to other treatment results, there was no significant difference among the other nitrogen forms whether UV-A was added or not, except the NH_4_^+^-N treatment with UV-A, which was higher than that without UV-A. The trend of ammonium nitrogen (NH_4_^+^-N) content was NH_4_^+^-N > NO_3_^−^-N/NH_4_^+^-N (3/1) > NO_3_^−^-N; however, after adding 20 µmol m^−2^ s^−1^ UV-A, there was no significant effect on the content of NH_4_^+^-N (Figure 4).

## 3. Discussion

Based on the ratio of ammonium nitrate, we added 20 µmol m^−2^ s^−1^ UV-A to increase the growth, photosynthesis rate, chlorophyll, and anthocyanin content by using a NO_3_^−^-N/NH_4_^+^-N (3/1) treatment on purple lettuce; this significantly reduced the nitrate content. The addition of 20 µmol·m^−2^·s^−1^ UV-A also significantly improved the nitrogen assimilation ability of each nitrogen form treatment, and thus increased the content of soluble protein and free amino acids in purple lettuce.

### 3.1. Adding 20 µmol m^−2^ s^−1^ UV-A to Different Ratios of Ammonium Nitrate Is Beneficial for the Improvement of Photosynthesis and Growth of Purple Leaf Lettuce

The two primary inorganic nitrogen types absorbed during plant development are nitrate nitrogen and ammonium nitrogen [8]. Studies have shown that a suitable ammonium and nitrate nitrogen combination promotes plant growth and development more than either a nitrate or an ammonium nitrogen alone, increasing vegetable production and quality [22,23]. However, by combining different intensities of UV-A with varied white, red, and blue light qualities, the ideal ratio of ammonium nitrate needed for plant development is also diverse. In this study, the NO_3_^−^-N/NH_4_^+^-N (3/1) treatment generated the maximum production of purple lettuce grown hydroponically, whereas the NH_4_^+^-N treatment produced the least yield, leaf area, and number of single leaves no matter the UV-A supplementation. The growth and development of hydroponic purple lettuce is supported by appropriately raising the nitrate nitrogen to nitrogen ratio and decreasing the ammonium nitrogen to nitrogen ratio. This is primarily due to the fact that nitrate nitrogen is the most readily assimilated nitrogen nutrient in some plants, and it can facilitate plant photosynthetic carbon assimilation and sucrose accumulation more than ammonium nitrogen [24]. When individual ammonium nitrogen is supplied to plants, they will produce a certain amount of ammonium poisoning, which will severely inhibit the growth of plants and have the opposite effect [25,26]. After adding UV-A, the yield is higher than that without UV-A, which indicates that the addition of 20 µmol m^−2^ s^−1^ UV-A to the NO_3_^−^-N/NH_4_^+^-N (3/1) is more conducive to the hydroponic purple lettuce yield accumulation.

The metabolism of all substances in nature is based on photosynthesis, including the uptake, transmission, and conversion of photosynthetic pigments, which are the building blocks of photosynthesis in plants [27]. Crops’ ability to produce photosynthetic organs, photosynthesis reactions, and gas exchange factors related to photosynthetic rate are all impacted by the nitrogen form. The improvement of plant photosynthesis is facilitated by an appropriate nitrogen ratio [28]. This study demonstrated that the interaction of various nitrogen forms and UV-A significantly affected the photosynthesis of hydroponic purple lettuce. The ratio of the NO_3_^−^-N/NH_4_^+^-N (3/1) treatment can promote the accumulation of chlorophyll and carotenoid content in hydroponic purple lettuce more than other nitrogen forms (Table 2), a finding that is consistent with that of Barickman’s study [29]. However, the content of chlorophyll after the addition of UV-A is higher than that without UV-A, indicating that UV-A can promote the content of chlorophyll in hydroponic purple lettuce under the NO_3_^−^-N/NH_4_^+^-N (3/1) treatment, and then promote purple lettuce photosynthetic capacity to improve its level of growth and development. When the ammonium-nitrate ratio was (3:1), the net photosynthetic rate of red pepper increased first and then decreased [30]. This study showed that, compared with a NO_3_^−^-N treatment, its net photosynthetic rate, intercellular CO_2_ concentration, transpiration rate, and stomatal conductance were all highest under the NO_3_^−^-N/NH_4_^+^-N (3/1) treatment (Figure 1). The photosynthetic rate of cucumber leaves under total nitrate treatment was significantly reduced, and the reason for this is that a high concentration of nitrate nitrogen treatment destroyed the basal layer of cucumber chloroplasts, which was not conducive to the photosynthesis. After adding UV-A, its stomatal conductance, intercellular CO_2_ concentration, and transpiration rate were all highest under the treatment of NO_3_^−^-N/NH_4_^+^-N (3/1), this shows that the addition of appropriate UV-A on the basis of ammonium nitrate application can promote stomata opening, improve photosynthetic efficiency, and thus promote yield formation. *Φ*_PSII_ is used to represent the quantum yield of plant photosynthetic electron transfer, which reflects the actual original light energy capture rate when the reaction center is partially closed. The results of this study indicate that the interaction between different nitrogen forms and UV-A creates a large difference in the fluorescence characteristics of hydroponic purple lettuce leaves (Table 3). From the perspective of different nitrogen forms, *Φ*_PSII_, ETR and *F*_v_/*F*_m_ were higher than those treated with total ammonium or total nitrate under the ratio of NO_3_^−^-N/NH_4_^+^-N (3/1). The *F*_v_/*F*_m_ reflects the maximum light energy conversion efficiency of the PSII reaction center. If the *F*_v_/*F*_m_ is greatly reduced, it indicates that the leaves have been light-suppressed [31]. After the addition of UV-A, the chlorophyll fluorescence parameters of hydroponic purple lettuce leaves showed a downward trend in various nitrogen forms. The reason for this may be that the hydroponic purple lettuce leaves were suppressed by ultraviolet light; this is consistent with the decreasing chloroplast fluorescence characteristics of rice after ultraviolet radiation [32]. Furthermore, phytohormones like auxins are a class of trace endogenic hormones that are essential for plant development and responses to biotic and abiotic stress [33]. The positive effects of UV-A supplementation on lettuce growth and photosynthesis were also related to hormone levels and tradeoffs, but this needs to be further investigated.

### 3.2. Adding 20 µmol m^−2^ s^−1^ UV-A Significantly Reduced the Nitrate Content of NO_3_^−^-N/NH_4_^+^-N (3/1)-Treated Purple Lettuce and Increased Its Anthocyanin Content

Vegetables are plants whose nitrate content can be easily increased. Nitrate levels have a negative association with vegetable quality and can adversely impair human health when they are too high. Numerous studies have demonstrated that the nitrate content of vegetables is significantly influenced by the amount of nitrogen fertilizer used [8,34]. This study demonstrated that the addition of nitrate nitrogen considerably raised the nitrate content in hydroponically grown purple lettuce, with the NO_3_^−^-N treatment having the greatest levels under the different nitrogen forms. The nitrate content steadily dropped when ammonium nitrate was combined with the lowest treatment, with NH_4_^+^-N being followed by the treatment with NO_3_^−^-N/NH_4_^+^-N (3/1) (Table 4). This is mainly due to the fact that nitrate nitrogen must first be reduced in order to form ammonium nitrogen before it can be used, which makes its accumulation in vegetables easier. Ammonium nitrogen, on the other hand, can directly contribute to the synthesis of organic matter that contains nitrogen after being absorbed by vegetables. The fact that the nitrate content is lower after UV-A addition than it was before shows that UV-A is helping to lower nitrate levels. The main source of nitrogen for plants is soluble protein. In this study, the hydroponic purple lettuce treated with NO_3_^−^-N/NH_4_^+^-N (3/1) had a higher soluble protein content than the NO_3_^−^-N treatment regardless of whether UV-A was added (Table 4). This is primarily because ammonium nitrogen can directly participate in the synthesis of soluble proteins, while nitrate nitrogen is indirectly involved in the synthesis of soluble proteins. Soluble protein showed a declining trend with an increase in nitrate nitrogen proportion, and its level is at its maximum under total ammonium [22]. After adding UV-A, its soluble protein increased to a certain extent, which shows that adding UV-A can promote protein synthesis under different nitrogen forms [35]. This experimental study found that, from the perspective of different nitrogen forms, compared with other treatments, ammonium nitrogen is more conducive to the synthesis of soluble sugar in hydroponic purple lettuce; it is the highest under the NH_4_^+^-N treatment, followed by the NO_3_^−^-N/NH_4_^+^-N (3/1) treatment. This is similar to the conclusion that ammonium nitrogen fertilizer is more beneficial for increasing the soluble sugar content in spinach than nitrate nitrogen fertilizer. The content of free amino acids and VC were highest under the treatment of NO_3_^−^-N/NH_4_^+^-N (3/1), and the lowest under the treatment of total nitrates. This is mainly because ammonium nitrogen can directly participate in the conversion and synthesis of amino acids, while nitrate nitrogen is converted into ammonium nitrogen by nitrate reductase (NR), and nitrite reductase (NiR) to synthesize amino acids. After adding UV-A, it is higher than without UV-A as a whole, which shows that UV-A can promote the content of free amino acids and VC in nitrogen form.

UV radiation affects the synthesis of secondary metabolism in plants [36]. Nitrogen nutrition is one of the important environmental factors affecting plant growth and secondary metabolic processes. It is generally believed that nitrogen nutrition will promote the synthesis of nitrogen-containing secondary metabolites, which reflects the carbon-nutritional balance requirements of plants [37]. In this experiment, the anthocyanins, total phenols, and flavonoids of each nitrogen form added with UV-A were higher than those without UV-A (Figure 2). This was especially true in terms of anthocyanin content, which is the same as UV-A [38], which can promote the anthocyanin content in grapes and lettuce. From different nitrogen forms, the total phenol and flavonoid content were highest under the treatment of NH_4_^+^-N and NO_3_^−^-N/NH_4_^+^-N (3/1); this is mainly because ammonium nitrogen is beneficial for the accumulation of photosynthetic products, such as soluble sugar, and the original source of flavonoids in plants is the photosynthetic product.

### 3.3. On the Basis of the Ratio of Ammonium Nitrate, the Addition of 20 µmol m^−2^ s^−1^ UV-A Significantly Improved the Nitrogen Assimilation Ability of Each Nitrogen Form Treatment

The assimilation and reduction of inorganic nitrogen as well as the production of organic molecules containing nitrogen are the primary tasks of nitrogen metabolism, which is crucial to plant activity. Among these, nitrate reductase (NR) is the initial enzyme in the reduction process of NO_3_^−^-N, as well as the rate-limiting and inducing enzyme in this process, and the activity of this enzyme directly influences how plants absorb and use nitrate nitrogen [39]. The findings of this study demonstrate that the treatment of each nitrogen form with 20 µmol m^−2^ s^−1^ UV-A is higher than those without UV-A, but the trends of each nitrogen form are similar, and the nitrate nitrogen content is lowest under the treatment of NH_4_^+^-N (Figure 3). This indicates that the treatment of NO_3_^−^-N and NO_3_^−^-N/NH_4_^+^-N (3/1) can promote the absorption and assimilation abilities of nitrate nitrogen in the leaves of purple lettuce. The research results of Wang et al. indicated that the enhanced UV-B+UV-A radiation increased the activity of NR, protease, and peptidase and, at the same time, increased the content of free amino acids and soluble proteins. This shows that plants accelerate nitrogen metabolism by accelerating protein synthesis and degradation to adapt to ultraviolet radiation to reduce damage to plants. This is the same as the increased trend of soluble protein and free amino acid content after adding 20 µmol m^−2^ s^−1^ UV-A in this experiment (Table 4). The results of this study showed that the NR and NiR activities of each nitrogen form treatment after the addition of 20 µmol m^−2^ s^−1^ UV-A significantly increased (Figure 3), indicating that plants may prevent UV-A damage by accelerating nitrogen metabolism. This is consistent with the results of Kumar et al., who found that, under UV-B radiation, the NR activity and GS activity of cyanobacterium increased [40]. Studies have shown that NR activity is significantly increased at the level of NO_3_^−^ [41]. The results of this experiment show that the NR activity is highest under the treatment of NH_4_^+^-N (Figure 3b), and a total ammonium treatment is more beneficial for increasing the activity of nitrate reductase [42]. This may be because the high concentration of NH_4_^+^ will cause ammonium poisoning, which affects the absorption of NH_4_^+^ by purple lettuce. In order to supply nitrogen to plants in a timely manner, NO_3_^−^ in the root is preferentially transported to the leaves through the xylem, resulting in enhanced NR activity in the leaves.

Glutamine synthetase (GS) is a key enzyme in the reduction process of NH_4_^+^-N and also a key enzyme linking nitrogen assimilation metabolism and inorganic metabolism. Glutamate synthase (GOGAT) is the rate-limiting enzyme in the process of NH_4_^+^ assimilation; glutamate dehydrogenase (GDH) can catalyze the synthesis and catabolism of glutamic acid and is also a key enzyme in the process of nitrogen metabolism [43]. The results of this study show that GS activity is higher under the treatment of NH_4_^+^-N regardless of whether UV-A is added, followed by the NO_3_^−^-N/NH_4_^+^-N (3/1) treatment, and the activity of GOGAT is also the highest under the NO_3_^−^-N/NH_4_^+^-N (3/1) treatment (Figure 4). This may be because GS at the center of ammonia assimilation performs the functions of synthetase and invertase. The increase in GS activity will promote the enhancement of nitrogen metabolism, so it leads to higher GS activity under the treatment of total ammonium. The GS activity of the leek’s leaves or roots is higher with a single NH_4_^+^-N treatment and a mixed application of ammonium nitrate than it is under nitrate nitrogen. While not the primary mechanism, an increase in the concentration of NH_4_^+^ in the medium accelerates the assimilation of plants to N by increasing the activity of GDH. At the same time, GDH activity was maximum under the treatment of NH_4_^+^-N and NO_3_^−^-N/NH_4_^+^-N (3/1) after adding 20 µmol m^−2^ s^−1^ UV-A. According to research by Srivastava and Singh, excessive ammonium concentrations might cause an increase in GDH activity, and it is hypothesized that GDH may play a part in reducing NH_4_^+^ plant toxicity [44]. It shows that the increase in GDH helps the metabolism of nitrogen, and the products of nitrogen metabolism can provide prerequisites for other substances with the ability to resist UV-A stress.

## 4. Materials and Methods

### 4.1. Growing Media and Seeding

The experiment was carried out in the phytotron at the Science and Technology Innovation Park of Shandong Agricultural University from March to June 2018. The seeds were sown in plastic trays (54 × 28 × 5 cm) containing a mixture of peat (meagre trophic peat type, PH = 6.0, Pindstrup Horticulture Co., Ltd., Copenhagen, Denmark), perlite and vermiculite (Youshun Environmental Protection Technology Ltd., Shandong, China) (3:1:1, *v*/*v*/*v*), which were subsequently placed in a greenhouse. We selected consistent growth seedlings when they had 4 true leaves and placed them in plastic pots with an inner diameter of 37.5 cm, a width of 29 cm, and a height of 12 cm for hydroponics growth. Each pot held 5 plants. The solution (PH = 6.0, EC = 2.0) was prepared using the Hoagland formula (Table 5) and exchanged every five days. 

### 4.2. Experimental Design and Growth Environment

The experiment used randomized complete blocks design, and the plant having 6 true leaves was transferred to the artificial climate chamber for treatments (Qiushi, Zhejiang, China). The LED intelligent light control console (UH-TGT300B, Unihero, Guangzhou, China) used as a light source contains red light (650~660 nm), blue light (450~460 nm), ultraviolet light (380~400 nm), and white light; the light beads of each light quality are evenly distributed, and the light intensity and photoperiod can be separately regulated. The photosynthetic photon flux density (PPFD) was 300 μmol m^−2^ s^−1^, and the PPFD was measured using a luminometer (3415FX, Spectrum Technologies, Inc. Aurora, CO, USA); the PPFD under different light-quality treatments is shown in Table 6. There were at least four pots and twenty plants per treatment. The environmental conditions of the phytotron were as follows: day temperature (25 ± 1) °C, night temperature (18 ± 1) °C, air relative humidity 70%~80%, photoperiod 12 h·d^−1^, and the CO_2_ content in the chamber matched that outside.

### 4.3. Growth and Photosynthesis Parameters

After 20 days of treatment, the samples were taken using the 5th leaf from the heart to the outside, and the physiological indexes were measured. A ruler was used to calculate the length of the stem from the root to the growth point. Using a caliper, the stem’s base was measured for diameter. A leaf area meter, model CI-202 (CID Bio-Science Inc., Camas, WA, USA), was used to calculate the size of the leaves. An electronic scale was used to weigh the fresh mass of the leaves, and the samples were dried at 105 °C for 2 h and 85 °C in an oven until they reached a constant dry weight.

The CIRAS-3 portable photosynthesis equipment (PP Systems, Amesbury, MA, USA) was used to measure the leaf photosynthesis parameters after 20 days of routine management. The fourth or fifth leaf, counted away from the heart, was chosen. A CO_2_ injection system provides CO_2_ at a concentration of 400 ± 10 μmol·mol^−1^, and the built-in LED light source offers a light intensity of 1000 μmol m^−2^ s^−1^. The FMS-2 portable modulated chlorophyll fluorometer (Hansatech, Norfolk, UK) was used to measure the parameters of chlorophyll fluorescence. The absorbance of ethanol extraction was measured at 645, 663, and 440 nm using a UV-spectrophotometer (Shimadzu UV-16A, Shimadzu Corporation, Kyoto, Japan) to determine the concentrations of chlorophyll and carotenoids [18]. 

### 4.4. Determination of Quality

The soluble sugar content was determined by the anthrone method [45]; soluble protein content was determined by Coomassie brilliant blue method [46]; Vitamin C content was determined by xylene extraction colorimetry [47]; free amino acid content was determined by ninhydrin colorimetry [48]; nitrate content was determined by salicylic acid colorimetry [49]. To measure the anthocyanin content, weigh 0.2 g of the shredded tissue leaves precisely, add 10 mL of a 2% hydrochloric acid/methanol solution, and soak the eye tissue for 2 h at room temperature in the dark until it is totally whitened and filtered. The volume was converted to a 50 mL volumetric flask using a 2% hydrochloric acid methanol solution, and the absorbance at 530 nm was measured to determine the anthocyanin content. Flavonoids and total phenol content were determined using kits produced by Suzhou Keming Biotechnology Co., Ltd. (Suzhou, China).

### 4.5. Nitrogen Metabolism-Related Enzymes

Utilizing the salicylic acid technique, NO_3_^−^-N was determined [50]. The activity of nitrate reductase (NR) was assessed using p-aminobenzenesulfonic acid colorimetry. To assess the nitrite reductase (NiR) activity of the absorbance at 540 nm and the NO_2_^−^ micrograms consumed per gram of sample per hour, refer to the kit’s instructions (Comin Biotechnology Co., Ltd., Suzhou, China). The NH_4_^+^-N concentration is evaluated using the phenol-hypochlorite method [50]. The activity of glutamine synthetase (GS) was assayed by the method of Liu and Kao, and the mixture contained in a final 1 mL and the absorbance of supernatant was read at 540 nm [51]. For glutamate dehydrogenase (GDH), glutamate synthase (GOGAT) activity is evaluated by measuring the absorbance at 340 nm and measured by the enzymatic oxidation of NADH at the absorbance of 340 nm [52].

### 4.6. Data Processing

All experiments acquired three biological replicates. Data were processed using Microsoft Excel 2007 and DPS 7.05 software. The effects of UV-A supplementation and the formation of nitrogen, as well as their interactions, were compared by using an analysis of variance followed by Duncan’s multiple range test using SPSS 20.0 (IBM, New York, NY, USA) at *p* < 0.05 level. Differences were considered to be statistically significant for *p*-values below 0.05. OriginPro 10.0 software was used for plotting.

## 5. Conclusions

In conclusion, after adding 20 µmol m^−2^ s^−1^ UV-A, the NO_3_^−^-N/NH_4_^+^-N treatment of 3/1 treatment dramatically decreased the nitrate content while increasing growth, photosynthetic rate, chlorophyll, carotenoid, and anthocyanin content in purple lettuce. Additionally, it improves the nitrogen assimilation ability of purple lettuce leaves, accelerates the operation of nitrogen metabolism in plants, and thus increases the content of soluble proteins and free amino acids in plants, while also promoting the accumulation of biomass. As a result, to improve the growth of purple lettuce in future plant factories, based on the NO_3_^−^-N/NH_4_^+^-N (3/1) treatment, adding 20 µmol m^−2^ s^−1^ UV-A is conducive to increasing the production of purple lettuce and reducing the nitrate content of the lettuce.

## Figures and Tables

**Figure 1 ijms-24-16791-f001:**
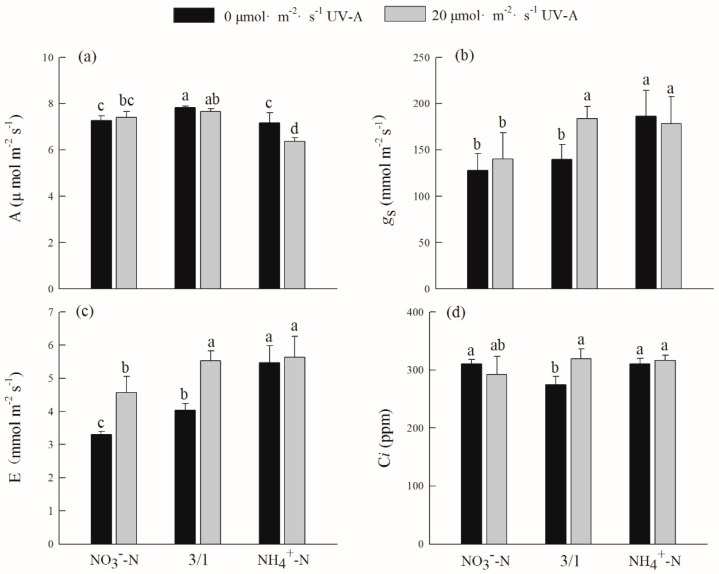
Effects of different nitrogen forms and UV-A interactions on gas exchange parameters of purple lettuce. (**a**) Net photosynthetic rate (A), (**b**) stomatal conductance (*g*s), (**c**) transpiration rate (E), (**d**) intercellular carbon dioxide concentration (*C*i). Different lower-case letters in the same column indicate a significant difference among treatments (Duncan’s multiple range test, *p* < 0.05). Values are means ± SD (*n* = 3).

**Figure 2 ijms-24-16791-f002:**
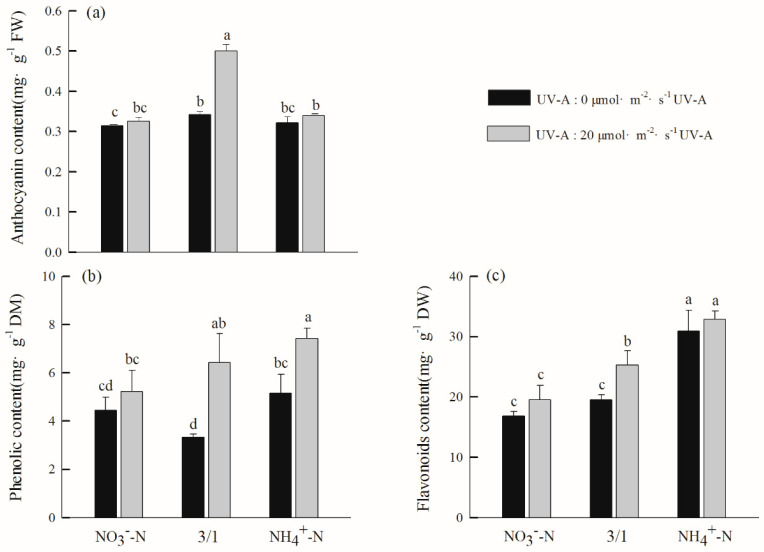
Effects of different nitrogen forms and UV-A interactions on secondary metabolites of purple lettuce: (**a**) anthocyanin, (**b**) total phenol, (**c**) flavonoids. Different lower-case letters in the same column indicate a significant difference among treatments (Duncan’s multiple range test, *p* < 0.05). Values are means ± SD (*n* = 3).

**Figure 3 ijms-24-16791-f003:**
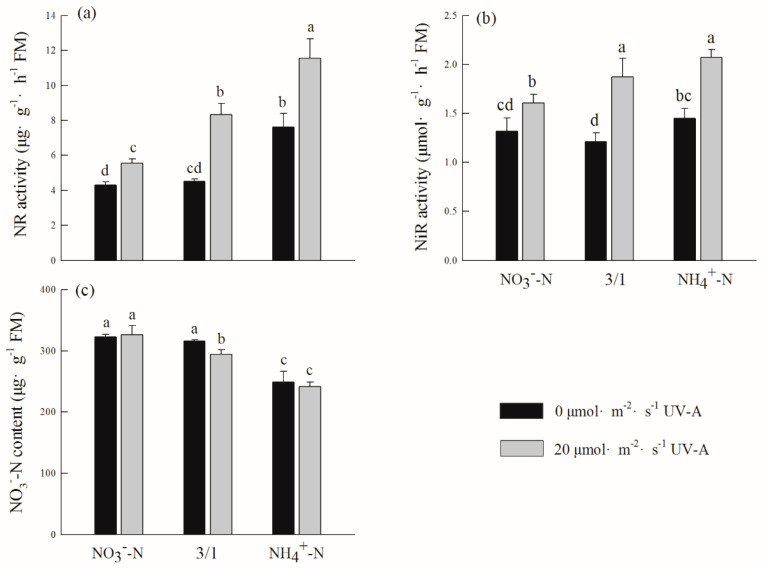
Effects of different ratios of ammonium nitrate and UV-A on nitrate nitrogen content and its reduction in purple lettuce: (**a**) nitrate reductase activity (NR), (**b**) nitrite reductase activity (NiR), (**c**) nitrate nitrogen content (NO_3_^−^-N). Different lower-case letters in the same column indicate a significant difference among treatments (Duncan’s multiple range test, *p* < 0.05). Values are means ± SD (*n* = 3).

**Figure 4 ijms-24-16791-f004:**
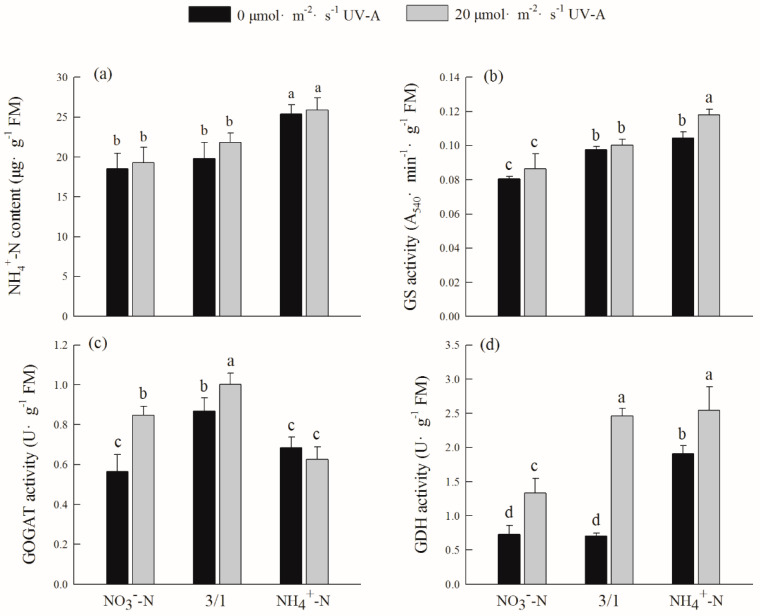
Effects of different ratios of ammonium nitrate and UV-A on ammonium nitrogen content and assimilation of purple lettuce: (**a**) ammonium nitrogen content (NH_4_^+^-N), (**b**) glutamine synthetase activity (GS), (**c**) glutamate synthase activity (GOGAT), (**d**) glutamate dehydrogenase activity (GDH). Different lower-case letters in the same column indicate a significant difference among treatments (Duncan’s multiple range test, *p* < 0.05). Values are means ± SD (*n* = 3).

**Table 1 ijms-24-16791-t001:** Effects of different nitrogen forms and UV-A interaction on the growth of purple lettuce.

UV-A	NO_3_^−^/NH_4_^+^	Stem Length (cm)	Stem Diameter (mm)	Leaf Area (cm^2^)	Fresh Weight (g)	Dry Weight (g)
0	NO_3_^−^-N	8.53 ± 0.99 b	1.096 ± 0.089 a	1727.61 ± 116.03 b	75.98 ± 6.29 b	3.68 ± 0.45 c
	3/1	7.56 ± 0.23 b	1.127 ± 0.023 a	2220.05 ± 320.04 a	84.76 ± 10.66 b	4.17 ± 0.5 bc
	NH_4_^+^-N	2.94 ± 0.74 c	0.85 ± 0.054 b	482.40 ± 23.82 c	15.49 ± 0.07 c	1.37 ± 0.06 d
	NO_3_^−^-N	11.05 ± 0.69 a	1.065 ± 0.084 a	2423.76 ± 22.39 a	98.43 ± 7.31 a	4.62 ± 0.38 ab
20	3/1	10.93 ± 0.68 a	1.071 ± 0.045 a	2494.38 ± 379.57 a	100.71 ± 5.99 a	5.06 ± 0.72 a
	NH_4_^+^-N	4.19 ± 0.17 c	0.938 ± 0.068 b	744.33 ± 124.74 c	20.83 ± 1.46 c	1.84 ± 0.17 d
Effects	UV-A	***	NS	**	***	**
	N	***	***	***	***	***
	UV-A × N	*	NS	NS	NS	NS

Different lower-case letters in the same column indicate a significant difference among treatments (Duncan’s multiple range test, *p* < 0.05). The symbol “*”, “**”, “***” and “NS” indicate significant level at *p* < 0.05, 0.01, 0.001 and no significance at *p* < 0.05, respectively. Values are means ± SD (*n* = 3).

**Table 2 ijms-24-16791-t002:** Effects of different nitrogen forms and UV-A interactions on photosynthetic pigment content of purple lettuce.

UV-A	NO_3_^−^/NH_4_^+^	Chlorophyll a (mg·g^−1^ FM)	Chlorophyll b (mg·g^−1^ FM)	Chlorophyll a + b (mg·g^−1^ FM)	Chlorophyll a/b (mg·g^−1^ FM)	Carotenoids (mg·g^−1^ FM)
0	NO_3_^−^-N	1.093 ± 0.219 e	0.318 ± 0.099 c	1.411 ± 0.318 e	3.515 ± 0.367 a	0.157 ± 0.024 b
	3/1	3.646 ± 0.341 ab	1.232 ± 0.085 ab	4.878 ± 0.388 ab	2.964 ± 0.263 ab	0.712 ± 0.114 a
	NH_4_^+^-N	1.481 ± 0.741 de	0.635 ± 0.503 c	2.116 ± 1.244 de	2.701 ± 0.738 b	0.238 ± 0.179 b
	NO_3_^−^-N	2.208 ± 0.624 cd	0.759 ± 0.143 bc	2.968 ± 0.766 cd	2.877 ± 0.263 ab	0.496 ± 0.121 ab
20	3/1	4.592 ± 0.629 a	1.605 ± 0.403 a	6.197 ± 0.959 a	2.952 ± 0.591 ab	0.733 ± 0.390 a
	NH_4_^+^-N	2.821 ± 0.387 bc	1.158 ± 0.229 ab	3.979 ± 0.519 bc	2.478 ± 0.428 b	0.465 ± 0.246 ab
	UV-A	**	**	**	NS	NS
Effects	N	***	**	***	NS	*
	UV-A × N	NS	NS	NS	NS	NS

Different lower-case letters in the same column indicate a significant difference among treatments (Duncan’s multiple range test, *p* < 0.05). The symbol “*”, “**”, “***” and “NS” indicate significant level at *p* < 0.05, 0.01, 0.001 and no significance at *p* < 0.05, respectively. Values are means ± SD (*n* = 3).

**Table 3 ijms-24-16791-t003:** Effects of different nitrogen forms and UV-A interactions on the chlorophyll fluorescence parameters of purple lettuce leaves.

UV-A	NO_3_^−^/NH_4_^+^	*F*_v_/*F*_m_	*F*_v_’/*F*_m_’	*Φ* _PSII_	*q*P	NPQ	ETR
0	NO_3_^−^-N	0.874 ± 0.03 ab	0.790 ± 0.03 a	0.598 ± 0.09 bc	0.756 ± 0.105 b	0.499 ± 0.29 bc	198.71 ± 31.64 bc
	3/1	0.891 ± 0.01 a	0.810 ± 0.01 a	0.718 ± 0.01 a	0.886 ± 0.03 a	0.710 ± 0.18 ab	238.25 ± 3.46 a
	NH_4_^+^-N	0.877 ± 0.01 ab	0.826 ± 0.03 a	0.683 ± 0.02 ab	0.828 ± 0.03 ab	1.001 ± 0.24 a	226.87 ± 6.51 ab
	NO_3_^−^-N	0.855 ± 0.01 bc	0.809 ± 0.01 a	0.520 ± 0.03 cd	0.643 ± 0.03 c	0.492 ± 0.16 bc	172.75 ± 9.88 cd
20	3/1	0.853 ± 0.01 bc	0.792 ± 0.02 a	0.589 ± 0.02 c	0.743 ± 0.01 b	0.465 ± 0.13 bc	195.44 ± 7.51 c
	NH_4_^+^-N	0.838 ± 0.02 c	0.791 ± 0.02 a	0.475 ± 0.05 d	0.599 ± 0.05 c	0.183 ± 0.13 c	157.62 ± 16.32 d
Effects	UV-A	**	NS	***	***	**	***
	N	NS	NS	*	**	NS	*
	UV-A × N	NS	NS	NS	NS	*	NS

Different lower-case letters in the same column indicate a significant difference among treatments (Duncan’s multiple range test, *p* < 0.05). The symbol “*”, “**”, “***” and “NS” indicate significant level at *p* < 0.05, 0.01, 0.001 and no significance at *p* < 0.05, respectively. Values are means ± SD (*n* = 3).

**Table 4 ijms-24-16791-t004:** Effects of different nitrogen forms and UV-A interactions on quality of purple lettuce.

UV-A	NO_3_^−^/NH_4_^+^	Soluble Sugar (mg·g^−1^ FM)	Soluble Protein (mg·g^−1^ FM)	Free Amino Acid (mg·g^−1^ FM)	Vc (mg·g^−1^ FM)	Nitrate (mg·g^−1^ FM)
0	NO_3_^−^-N	5.237 ± 0.48 b	5.073 ± 0.24 b	0.154 ± 0.004 c	1.578 ± 0.01 c	0.252 ± 0.03 a
	3/1	6.395 ± 0.22 a	5.159 ± 0.34 b	0.229 ± 0.026 a	1.612 ± 0.02 b	0.214 ± 0.02 b
	NO_3_^−^-N	6.621 ± 0.23 a	5.769 ± 0.05 a	0.226 ± 0.017 a	1.596 ± 0.02 bc	0.199 ± 0.01 bc
20	NO_3_^−^-N	6.273 ± 0.13 a	5.088 ± 0.05 b	0.189 ± 0.008 b	1.620 ± 0.02 b	0.219 ± 0.02 b
	3/1	6.282 ± 0.20 a	5.198 ± 0.11 b	0.230 ± 0.001 a	1.677 ± 0.01 a	0.178 ± 0.01 c
	NO_3_^−^-N	6.687 ± 0.25 a	5.759 ± 0.08 a	0.220 ± 0.007 a	1.649 ± 0.01 a	0.171 ± 0.02 c

Different lower-case letters in the same column indicate a significant difference among treatments (Duncan’s multiple range test, *p* < 0.05). Values are means ± SD (*n* = 3).

**Table 5 ijms-24-16791-t005:** Nutrient solution composition (mM) at a constant N concentration (7 mM) and different NO_3_^−^/NH_4_
^+^ ratios.

Nutrient Source	NO_3_^−^/NH_4_^+^		
	NO_3_^−^	NO_3_^−^/NH_4_^+^ = 3:1	NH_4_^+^
KNO_3_	3	0	0
Ca(NO_3_)_2_	2	2.625	0
KH_2_PO_4_	1	0	0
NH_4_H_2_PO_4_	0	1	1
MgSO_4_	1	1	1
K_2_SO_4_	0	2	2
CaCl_2_	0.625	0	2.625
(NH_4_)_2_SO_4_	0	0.375	3

**Table 6 ijms-24-16791-t006:** Different light quality ratio and nitrogen form ratio.

Light Quality Ratio W:R:B:UV-A	NO_3_^−^/NH_4_^+^
150:120:30:0	NO_3_^−^-N
	NO_3_^−^-N/NH_4_^+^-N = 3/1
	NH_4_^+^-N
130:120:30:20	NO_3_^−^-N
	NO_3_^−^-N/NH_4_^+^-N = 3/1
	NH_4_^+^-N

W: white light; R: red light; B: blue light; UV-A: ultraviolet-A light.

## Data Availability

All data of this study are available from the corresponding author upon reasonable request.

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
