# Peer review of "Appropriate Nitrogen form Ratio and UV-A Supplementation Increased Quality and Production in Purple Lettuce (Lactuca sativa L.)"

_ijms, 2023, doi:10.3390/ijms242316791_

Round 1

Reviewer 1 Report

Comments and Suggestions for Authors

Dear Authors,

Please find my suggestions below.

Specific comments

 Materials and Methods

P2/L80-81. Please write the ratio of peat, perlite and vermiculite, and the trademark, country. For peat, please write type and pH.

P2/L84. Please write the pH and the electrical conductivity of the nutrient solution.

P3/L94. Do the authors mean 300 μmol m-2 sec-1?

P3/L98. Do the authors measure the CO2 in the chamber?

P3/L106. Why did the authors use 85 οC for drying? Is there any reference?

P3/L113-114. Please write more details for the application of the method described by Mao et al. (2021) in your experiment (e.g. measurement of absorbance at 645, 663 and 440 nm). Please write the number of plants. the number of leaves per plant (which leaves) and the time of measurement.

P3/L126. Please add a reference for the method.

P4/140-143. Please write the number of replicates per treatment, the number of plants per replicate. Did the authors use for each treatment only 5 plants cultivated in one pot and then took three of these plants for the measurements? In my opinion the number of plants is very small for agricultural experimentation even in controlled atmosphere. The small number replicates per treatment or/and the small number of plants per replicate affects negatively the reliability of the results.

In addition, the experiment is bifactorial, factor A: UV-A (with two levels) and factor B: nitrogen form (with three levels). The statistical analysis made by the authors seems to be incorrect because they do not refer anything for the interaction of the two factors which is present in some of the parameters. In the case of statistical interaction between two factors, the effect of each factor is being considered in each level of the other factor separately. Moreover, the statistical analysis of a bifactorial experiment as a one factor experiment with the application of one-way ANOVA may lead to incorrect results and conclusions.

Examples:

1. Table 3 – leaf area: the value 482.40±23.82 c (UV-A 0/NH4+-N) does not differ significantly with the value 744.33 c (UV-A 20/NH4+-N). According to the standard deviations, it seems that these values differ significantly. The authors can find it with a T-test, which is appropriate in the case of the bifactorial analysis, in order to find the effect of the factor UV-A (with two levels) in each of the three levels of the factor “nitrogen form” separately.

2. Table 3 – fresh weight: the value 15.49±0.07 c (UV-A 0/NH4+-N) does not differ significantly with the value 20.83±1.46 (UV-A 20/NH4+-N). According to the standard deviations, it seems that these values differ significantly.

3. Table 4 – carotenoids: the value 0.157±0.024 b (UV-A 0/NO3--N) does not differ significantly with the value 0.496±0.121 ab (UV-A 20/NO3--N). According to the standard deviations, it seems that these values differ significantly.

4. Table 4 – chlorophyll a+b: the value 4.878±0.388 a (UV-A 0/NO3--N) does not differ significantly with the value 6.197±0.959 ab (UV-A 20/NO3--N). According to the standard deviations, it seems that these values differ significantly.

5. Table 5 – Fv΄/Fm΄: the value 0.809±0.01 a (UV-A 20/NO3--N) does not differ significantly with the value 0.791±0.02 a (UV-A 20/NH4+-N). According to the standard deviations, it seems that these values differ significantly. The authors can find it with an ANOVA, which is appropriate in the case of the bifactorial analysis, in order to find the effect of the factor “nitrogen form” (with three levels) in each of the two levels of the factor “UV-A” separately.

Results

In some cases, the standard deviation is very high, e.g. chlorophyll a, chlorophyll b and carotenoids in treatment UV-A 0/NH4+-N. Is there any explanation? Does it affect the statistical analysis? Could this be due to the small number of plants for each treatment?

Discussion

P9/L296-297. The authors write that NO3--N/NH4+-N (3/1) treatment generated the maximum production. This seems to be incorrect because in Table 3 – fresh weight (and dry weight), the authors present that the treatment UV-A 0/NO3--N does not differ significantly from the treatment UV-A 0/NO3--N/NH4+-N (3/1) and the treatment UV-A 20/NO3--N does not differ significantly from the treatment UV-A 20/NO3--N/NH4+-N (3/1).

P10/L306. The authors write that the addition of UV-A leads to higher yield. This seems to be incorrect in the treatment with NH4+-N. The treatment UV-A 0/NH4+-N does not differ significantly from the treatment UV-A 20/NH4+-N. Please apply the appropriate statistical analysis, as I recommend above.

Please rewrite the results, the discussion and the conclusions section according to the findings after the application of the appropriate statistical analysis.

Comments on the Quality of English Language

Few corrections in English language are required. 

Author Response

Many thanks for handling our manuscript (ijms-2626544), and please now find enclosed a revised version of the manuscript, entitled: Appropriate nitrogen form ratio and UV-A supplementation increased quality and production in purple lettuce (Lactuca sativa L.). We are re-submitting for consideration to publication as a Research Article in International Journal of Molecular Sciences.

According to both editorial and the reviewers’ comments and suggestions, we have revised and adjusted the manuscript according to International Journal of Molecular Sciences style, modified the text and figures, rewrote the results and discussion, and also added additional detailed information in the revised version of the manuscript. Revised portion are marked in red in the manuscript. The main correction in the paper and responds to the editor and reviewer’s comments are as flowing:

Reviewer 1:

P2/L80-81. Please write the ratio of peat, perlite and vermiculite, and the trademark, country. For peat, please write type and pH.

R: Thanks for recognition of reviewers to our works, which has helped a lot in boosting our self-confidence and motivation. We have made a corresponding revision to your valuable comments. Thanks a lot, again.

The seeds were sown in plastic trays (54×28×5 cm) containing a mixture of peat (meagre trophic peat type, PH=6.0, Pindstrup, Denmark), perlite and vermiculite (Youshun Environmental Protection Technology Ltd., China) (3: 1: 1, v/v/v).

P2/L84. Please write the pH and the electrical conductivity of the nutrient solution.

R: The solution (PH=6.0, EC=2.0) is prepared using the Hoagland formula (Table 1), and exchanged per five days.  

P3/L94. Do the authors mean 300 μmol m-2 sec-1?

R: We apologize for our negligence. the PPFD was 300 μmol m-2 s-1.

P3/L98. Do the authors measure the CO2 in the chamber?

R: The CO2 content in the chamber is about 380 ppm.

P3/L106. Why did the authors use 85 οC for drying? Is there any reference?

R: The samples were dried at 105 °C for 2 hour and 85 °C in an oven until they reached a constant dry weight. We used 85 °C because we wanted to speed up the drying process, and it's common that it might be 80 °C . The references are as follows:

Cui, G.; Zhao, X.; Liu, S.; Sun, F.; Zhang, C.; Xi, Y. Beneficial Effects of Melatonin in Overcoming Drought Stress in Wheat Seedlings. Plant Physiology and Biochemistry 2017, 118, 138–149. https://doi.org/10.1016/j.plaphy.2017.06.014.

P3/L113-114. Please write more details for the application of the method described by Mao et al. (2021) in your experiment (e.g. measurement of absorbance at 645, 663 and 440 nm). Please write the number of plants. the number of leaves per plant (which leaves) and the time of measurement.

R: There are four pots and twenty plants at least under per treatments.

After 20 days of treatment, the samples were taken as the 5th leaf from the heart to the outside, and the physiological indexes were measured. Each treatment was repeated three times.

P3/L126. Please add a reference for the method.

R: A reference for the method added, and as follows:

Solórzano, L. Determination of Ammonia in Natural Waters by the Phenolhypochlorite Method. Limnol. Oceanogr. 1969,14 (5), 799–801. https://doi.org/10.4319/lo.1969.14.5.0799.

P4/140-143. Please write the number of replicates per treatment, the number of plants per replicate. Did the authors use for each treatment only 5 plants cultivated in one pot and then took three of these plants for the measurements? In my opinion the number of plants is very small for agricultural experimentation even in controlled atmosphere. The small number replicates per treatment or/and the small number of plants per replicate affects negatively the reliability of the results.

R: The experiment was performed in pot of artificial climate chamber, and each pot should hold 5 plants. There are four pots and twenty plants at least per treatments. The samples were taken as the 5th leaf from the heart to the outside, and the physiological indexes were measured. All experiment taken three biological replicates.

In addition, the experiment is bifactorial, factor A: UV-A (with two levels) and factor B: nitrogen form (with three levels). The statistical analysis made by the authors seems to be incorrect because they do not refer anything for the interaction of the two factors which is present in some of the parameters. In the case of statistical interaction between two factors, the effect of each factor is being considered in each level of the other factor separately. Moreover, the statistical analysis of a bifactorial experiment as a one factor experiment with the application of one-way ANOVA may lead to incorrect results and conclusions. 

Examples:

  1. Table 3 – leaf area: the value 482.40±23.82 c (UV-A 0/NH4+-N) does not differ significantly with the value 744.33 c (UV-A 20/NH4+-N). According to the standard deviations, it seems that these values differ significantly. The authors can find it with a T-test, which is appropriate in the case of the bifactorial analysis, in order to find the effect of the factor UV-A (with two levels) in each of the three levels of the factor “nitrogen form” separately.
  2. Table 3 – fresh weight: the value 15.49±0.07 c (UV-A 0/NH4+-N) does not differ significantly with the value 20.83±1.46 (UV-A 20/NH4+-N). According to the standard deviations, it seems that these values differ significantly.
  3. Table 4 – carotenoids: the value 0.157±0.024 b (UV-A 0/NO3--N) does not differ significantly with the value 0.496±0.121 ab (UV-A 20/NO3--N). According to the standard deviations, it seems that these values differ significantly.
  4. Table 4 – chlorophyll a+b: the value 4.878±0.388 a (UV-A 0/NO3--N) does not differ significantly with the value 6.197±0.959 ab (UV-A 20/NO3--N). According to the standard deviations, it seems that these values differ significantly.
  5. Table 5 –Fv΄/Fm΄: the value 0.809±0.01 a (UV-A 20/NO3--N) does not differ significantly with the value 0.791±0.02 a (UV-A 20/NH4+-N). According to the standard deviations, it seems that these values differ significantly. The authors can find it with an ANOVA, which is appropriate in the case of the bifactorial analysis, in order to find the effect of the factor “nitrogen form” (with three levels) in each of the two levels of the factor “UV-A” separately.

R: Thank you to the reviewers for their careful observation and insightful comments. We have verified the data and analyzed it using F-test. The results are shown in the table and picture below:

UV-A

NO3- / NH4+

Stem length (cm)

Stem diameter (mm)

Leaf area (cm2)

Fresh weight (g)

Dry weight (g)

0

NO3--N

8.53±0.99 b

1.096±0.089 a

1727.61±116.03 b

75.98±6.29 b

3.68±0.45 c

3/1

7.56±0.23 b

1.127±0.023 a

2220.05±320.04 a

84.76±10.66 b

4.17±0.5 bc

NH4+-N

2.94±0.74 c

0.85±0.054 b

482.40±23.82 c

15.49±0.07 c

1.37±0.06 d

NO3--N

11.05±0.69 a

1.065±0.084 a

2423.76±22.39 a

98.43±7.31 a

4.62±0.38 ab

20

3/1

10.93±0.68 a

1.071±0.045 a

2494.38±379.57 a

100.71±5.99 a

5.06±0.72 a

NH4+-N

4.19±0.17 c

0.938±0.068 b

744.33±124.74 c

20.83±1.46 c

1.84±0.17 d

Effects

UV-A

***

NS

**

***

**

N

***

***

***

***

***

UV-A * N

*

NS

NS

NS

NS

Results

In some cases, the standard deviation is very high, e.g. chlorophyll a, chlorophyll b and carotenoids in treatment UV-A 0/NH4+-N. Is there any explanation? Does it affect the statistical analysis? Could this be due to the small number of plants for each treatment?

R: Thanks for expert review to improve our paper. We examined our raw data and operational processes and found no problems, and the standard deviation is very high, most likely because the samples were not mixed during sampling. The experiment was performed in pot of artificial climate chamber, and each pot should hold 5 plants. There are four pots and twenty plants at least per treatments. The samples were taken as the 5th leaf from the heart to the outside, and the physiological indexes were measured. All experiment taken three biological replicates, and the data were processed using Microsoft Excel 2007 and DPS 7.05 software. Besides, the interaction effects of UV-A supplementation and nitrogen form were compared by analysis of variance followed by Duncan’s multiple range test using SPSS 20.0 (IBM, New York, USA) at P < 0.05 level.

Discussion

P9/L296-297. The authors write that NO3--N/NH4+-N (3/1) treatment generated the maximum production. This seems to be incorrect because in Table 3 – fresh weight (and dry weight), the authors present that the treatment UV-A 0/NO3--N does not differ significantly from the treatment UV-A 0/NO3--N/NH4+-N (3/1) and the treatment UV-A 20/NO3--N does not differ significantly from the treatment UV-A 20/NO3--N/NH4+-N (3/1). 

R: We also noticed what this issue mentioned reviewer. I am sorry that this part was not clear in the original manuscript. I should have explained that we would like to emphasize the positive role of NO3--N/NH4+-N (3/1) treatment in increasing yield no matter the UV-A supplementation and compare it with previous studies in the discussion section. Although there is no significant difference between UV-A 0/NO3--N and UV-A 0/NO3--N/NH4+-N (3/1) and UV-A 20/NO3--N and UV-A 20/NO3--N/NH4+-N (3/1) treatments in the Table 3, it is obvious that fresh weight (and dry weight) under UV-A 0/NO3--N/NH4+-N (3/1) and UV-A 20/NO3--N/NH4+-N (3/1) treatment is higher than UV-A 0/NO3--N  and UV-A 20/NO3--N treatment, respectively. I have revised the contents of this part.

P10/L306. The authors write that the addition of UV-A leads to higher yield. This seems to be incorrect in the treatment with NH4+-N. The treatment UV-A 0/NH4+-N does not differ significantly from the treatment UV-A 20/NH4+-N. Please apply the appropriate statistical analysis, as I recommend above.

R: Thank you to the reviewers for their suggestions! We have added the analysis of variance followed by Duncan’s multiple range test using SPSS 20.0. Although there is no significant difference between the treatment UV-A 0/NH4+-N and UV-A 20/NH4+-N, the addition of UV-A is effective in increasing yield after summarizing all the data.

Please rewrite the results, the discussion and the conclusions section according to the findings after the application of the appropriate statistical analysis.

R: We rewrite the results, the discussion and the conclusions section. See the manuscript for details.

Comments on the Quality of English Language

Few corrections in English language are required. 

R: We have checked the manuscript in its entirety for spelling errors, grammatical errors, and structural issues.

We tried our best to improve the manuscript and made some changes according to the editor and reviewer’s comments. We appreciate for editor and reviewer’s warm work earnestly, and hope that the correction will meet with approval. Thank you very much for your comments and suggestions, once again.

Looking forward to hearing from you.

With best wishes

Yours sincerely,

Dr. Yinjian Zheng

Professor,

Institute of Urban Agriculture

Chinese Academy of Agricultural Science

Reviewer 2 Report

Comments and Suggestions for Authors

The study is based on the impact of supply UVA to different form of the Nitrogen and their impact on Lactuca sativa growth. Authors have measured, under chamber conditions, biochemical, enzyamtic and growth parameters.

The manuscript is intersting the used methods are almost current and sound.

The results seems interesting and the discussion sometimes speculative

The conclusion are related to the objectives.

Nevertheless, several shortcomings are present.

Methodologically,

The experimental design is not presented. Moreover, the authors seem used randomized blocks. This is not adequate design. Two factors are studied here and the most judicious design is split plot one. Therefore, the statistical analyses should be done accordingly with factor (Nitrogen form, for example) and subfactor (UV-A supply). the interaction between trait is not displayed.

Scientifically

1-it is well known that plants prefer NO3 form for a better growth. This is truer for plants exploited for their vegetatve development like lettuce. Therefore the presented results did not bring high novelty.

2-the positif impact of UV-A is not presented. Instead, authors have presented speculative explanation on the role of UVA as stimulant of enzymes involved in nitrogen used and modification.

Why authors did not prospect, at least discuss, the effect on phytohormones ? It is known that UV raditions affect auxins functioning which may results in impact on vegetative growth. This point should be added in discussion (even authors did not assess phytohormones).

Minor remark

Authors should follow the journal guidelines for the indexation of the references in the text.

Comments on the Quality of English Language

The English needs minor corrections, modtly due to typo errors

Author Response

Many thanks for handling our manuscript (ijms-2626544), and please now find enclosed a revised version of the manuscript, entitled: Appropriate nitrogen form ratio and UV-A supplementation increased quality and production in purple lettuce (Lactuca sativa L.). We are re-submitting for consideration to publication as a Research Article in International Journal of Molecular Sciences.

According to both editorial and the reviewers’ comments and suggestions, we have revised and adjusted the manuscript according to International Journal of Molecular Sciences style, modified the text and figures, rewrote the results and discussion, and also added additional detailed information in the revised version of the manuscript. Revised portion are marked in red in the manuscript. The main correction in the paper and responds to the editor and reviewer’s comments are as flowing:

Reviewer 2:

Comments and Suggestions for Authors

The study is based on the impact of supply UVA to different form of the Nitrogen and their impact on Lactuca sativa growth. Authors have measured, under chamber conditions, biochemical, enzyamtic and growth parameters.

The manuscript is intersting the used methods are almost current and sound.

The results seems interesting and the discussion sometimes speculative

The conclusion are related to the objectives.

Nevertheless, several shortcomings are present.

R: We thank the reviewers for reviewed our manuscript on their busy schedules. We apologize for the delay in correcting the manuscript due to the short time available.

Methodologically, 

The experimental design is not presented. Moreover, the authors seem used randomized blocks. This is not adequate design. Two factors are studied here and the most judicious design is split plot one. Therefore, the statistical analyses should be done accordingly with factor (Nitrogen form, for example) and subfactor (UV-A supply). the interaction between trait is not displayed.

R: Thank you to the reviewers for their careful observation and insightful comments. We have verified the data and analyzed it using F-test. The results are shown in the table and picture below:

UV-A

NO3- / NH4+

Stem length (cm)

Stem diameter (mm)

Leaf area (cm2)

Fresh weight (g)

Dry weight (g)

0

NO3--N

8.53±0.99 b

1.096±0.089 a

1727.61±116.03 b

75.98±6.29 b

3.68±0.45 c

3/1

7.56±0.23 b

1.127±0.023 a

2220.05±320.04 a

84.76±10.66 b

4.17±0.5 bc

NH4+-N

2.94±0.74 c

0.85±0.054 b

482.40±23.82 c

15.49±0.07 c

1.37±0.06 d

NO3--N

11.05±0.69 a

1.065±0.084 a

2423.76±22.39 a

98.43±7.31 a

4.62±0.38 ab

20

3/1

10.93±0.68 a

1.071±0.045 a

2494.38±379.57 a

100.71±5.99 a

5.06±0.72 a

NH4+-N

4.19±0.17 c

0.938±0.068 b

744.33±124.74 c

20.83±1.46 c

1.84±0.17 d

Effects

UV-A

***

NS

**

***

**

N

***

***

***

***

***

UV-A * N

*

NS

NS

NS

NS

Scientifically

1-it is well known that plants prefer NO3 form for a better growth. This is truer for plants exploited for their vegetatve development like lettuce. Therefore the presented results did not bring high novelty.

2-the positif impact of UV-A is not presented. Instead, authors have presented speculative explanation on the role of UVA as stimulant of enzymes involved in nitrogen used and modification.

R: The experiment purpose was to increase the yield and quality of purple lettuce while lowering its nitrate level. By adding various ratios of NO3--N and NH4+-N to the nutrient solution and 20 µmol m-2 s-1 UV-A based on white, red, and blue light (130, 120, 30 µmol m-2 s-1), the effects of different NO3--N/NH4+-N ratios (NO3--N, NO3--N/NH4+-N=3/1, NH4+-N) and UV-A interaction on yield, quality, photosynthetic characteristics, anthocyanins, and nitrogen assimilation of purple lettuce were studied. We believe that this is a theoretical foundation and technological specifications in vegetable cultivation like lettuce.

Why authors did not prospect, at least discuss, the effect on phytohormones ? It is known that UV raditions affect auxins functioning which may results in impact on vegetative growth. This point should be added in discussion (even authors did not assess phytohormones).

R: Thank you very much for the reviewer's suggestion, we have added the relevant description in the discussion section.

Minor remark

Authors should follow the journal guidelines for the indexation of the references in the text.

      R: We have checked and corrected reference citations according to journal guidelines.

Comments on the Quality of English Language

The English needs minor corrections, modtly due to typo errors

R: We have checked the manuscript in its entirety for spelling errors, grammatical errors, and structural issues.

We tried our best to improve the manuscript and made some changes according to the editor and reviewer’s comments. We appreciate for editor and reviewer’s warm work earnestly, and hope that the correction will meet with approval. Thank you very much for your comments and suggestions, once again.

Looking forward to hearing from you.

With best wishes

Yours sincerely,

Dr. Yinjian Zheng

Professor,

Institute of Urban Agriculture

Chinese Academy of Agricultural Science

Round 2

Reviewer 2 Report

Comments and Suggestions for Authors

Dear authors,

thank you for your efforts to improve the manuscript.

Nevertheless, the important requested  changes have not been done.

The experimental design is not presneted . Did you used  randomized complete blocks design ? or a split plot one?

concerning the impact of UV-A on phytohormones, authors have added a couple of sentences without any connexion with the discussion and without references.

Comments on the Quality of English Language

minor modifications are necessary.

Usually, the past tense is used for the presentation of the results.

Author Response

Many thanks for handling our manuscript (ijms-2626544). According to both editorial and the reviewers’ comments and suggestions, we have revised and adjusted the manuscript in the revised manuscript. Revised portion are marked in red in the manuscript. The main correction in the paper and responds to the editor and reviewer’s comments are as flowing:

The experimental design is not presneted . Did you used  randomized complete blocks design ? or a split plot one?

The experiment used randomized complete blocks design, and the plant having 6 true leaves transferred to the artificial climate chamber for treatments.  The effect of various nitrogen form ratio and UV-A supplementation on yield, quality, photosynthetic characteristics, anthocyanins, and nitrogen assimilation of purple lettuce were studied in the artificial climate chamber.   concerning the impact of UV-A on phytohormones, authors have added a couple of sentences without any connexion with the discussion and without references.   Many thanks to the reviewers for their suggestions to enhance our manuscript! Light, including UV-A, can affect plant growth and development and metabolite synthesis in terms of both energy and signaling process, in which phytohormones play a crucial role. However, the purpose in this manuscript  was to increase the yield and quality of purple lettuce while lowering its nitrate level. We have explored the mechanisms in terms of photosynthesis and enzyme activity. In addition, based on the reviewers' comments, we have also described the possible role of phytohormones and added the relevant references in the discussion section. Because of the lack of experimental data, we cannot draw a conclusions from an objective point of view. We apologize for this.

We tried our best to improve the manuscript and made some changes according to the editor and reviewer’s comments. We appreciate for editor and reviewer’s warm work earnestly, and hope that the correction will meet with approval. Thank you very much for your comments and suggestions, once again.

Looking forward to hearing from you.

With best wishes!

Yours sincerely,

Yinjian Zheng

Professor,

Institute of Urban Agriculture

Chinese Academy of Agricultural Science

Round 3

Reviewer 2 Report

Comments and Suggestions for Authors

Dear authors,

Thank you for your efforts to modify deeply, appropriately and properly your manuscript. The answers to remarks were persuasive.